# Patterns and correlates of physical activity and sedentary behavior among Bangkok residents: A cross-sectional study

Thitikorn Topothai[1,2,3]*, Viroj Tangcharoensathien[2], Sarah Martine Edney[1], Rapeepong Suphanchaimat[2,4], Angkana Lekagul[2], Orratai Waleewong[2], Chompoonut Topothai[1,2,5], Anond Kulthanmanusorn[2], Falk Müller-Riemenschneider[1,6,7]

**1** Saw Swee Hock School of Public Health, National University of Singapore and National University Health System, Singapore, Singapore, **2** International Health Policy Program, Ministry of Public Health, Nonthaburi, Thailand, **3** Division of Physical Activity and Health, Department of Health, Ministry of Public Health, Nonthaburi, Thailand, **4** Division of Epidemiology, Department of Disease Control, Ministry of Public Health, Nonthaburi, Thailand, **5** Bureau of Health Promotion, Department of Health, Ministry of Public Health, Nonthaburi, Thailand, **6** Yong Loo Lin School of Medicine, National University of Singapore, Singapore, Singapore, **7** Digital Health Center, Berlin Institute of Health, Charité-Universitätsmedizin Berlin, Berlin, Germany

* thitikorn@u.nus.edu

**Data Availability Statement:** the data can be downloaded from the Thailand National Statistics Office's website: http://www.nso.go.th/sites/

## Abstract

### Background

Physical inactivity and sedentary behavior are significant risk factors for various non-communicable diseases. Bangkok, Thailand's capital, is one of the fastest-growing metropolitans in Southeast Asia. Few studies have investigated the epidemiology of physical activity and sedentary behavior among Bangkok residents. This study aims to investigate the prevalence of combined physical activity and sedentary behavior patterns among Bangkok residents and examine relationships between participants' characteristics and the combined movement patterns.

### Methods

We analyzed data from the nationally representative 2021 Health Behavior Survey conducted by the Thailand National Statistical Office. The Global Physical Activity Questionnaire was used to assess physical activity and sedentary behavior. 'Sufficiently active' was defined as meeting the World Health Organization's guidelines for aerobic physical activity ($\geq$150 minutes of moderate-to-vigorous physical activity per week). 'Low sedentary time' was defined as sitting for $\leq$7 hours per day. Participants were categorized into one of four movement patterns: highly active/low sedentary, highly active/highly sedentary, low active/low sedentary, and low active/highly sedentary. Multinomial logistic regression was used to identify the factors associated with each group of four movement patterns.

2014en/Pages/survey/Social/Health/The-2020-Health-behavior-population-survey.aspx.

**Funding:** (1) Thailand Science Research and Innovation (TSRI) for the Senior Research Scholar on Health Policy and System Research (Contract No. RTA6280007) and (2) the Capacity Building on Health Policy and Systems Research program (HPSR Fellowship) under cooperation between the Bank for Agriculture and Agricultural Co-operatives (BAAC), National Health Security Office (NHSO), and International Health Policy Program Foundation (IHPF).

**Competing interests:** The authors have declared that no competing interests exist.

**Abbreviations:** AOR, Adjusted odds ratio; BMI, Body mass index; CI, Confidence interval; GPAQ, Global Physical Activity Questionnaire; METs, Metabolic equivalents; NCD, Non-communicable diseases; NSO, National Statistical Office; WHO, World Health Organization.

## Results

Among the 3,137 individuals included in the study, the majority were categorized as highly active/highly sedentary (64.8%), followed by highly active/low sedentary (17.9%) and low active/highly sedentary (14.3%). Only a few (3.0%) of participants were categorized as being low active/low sedentary. Compared to males, female participants had a significantly higher likelihood of belonging to the highly active/low sedentary (AOR = 1.69, 95%CI: 1.25, 2.28) or highly active/highly sedentary (AOR = 1.51, 95%CI: 1.19, 1.93) group, rather than the low active/high sedentary group. Compared to unemployed/retired participants, those in labor-intensive occupations had a significantly higher likelihood of being in the highly active/ low sedentary group (AOR = 1.89, 95%CI: 1.22, 2.94). Compared to participants with no chronic physical conditions, participants who reported multimorbidity had a significantly lower likelihood of being in the highly active/low sedentary group (AOR = 0.60, 95%CI: 0.37, 0.98).

## Conclusion

This study provides valuable insights into the patterns of physical activity and sedentary behavior among residents of Bangkok using up-to-date data. The majority belonged to the highly active/highly sedentary group, followed by the highly active/low sedentary group. Correlates such as sex, occupation, and chronic conditions were associated with these patterns. Targeted interventions in recreational activities, workplaces, and urban areas, including screen time control measures, movement breaks and improved built environments, are crucial in reducing sedentary behavior and promoting physical activity.

## Background

Physical inactivity and sedentary behavior are well-established risk factors for non-communicable diseases (NCDs), including coronary heart disease, type 2 diabetes, dementia, depression, and premature mortality [1, 2]. The prevalence of physical inactivity globally is a concern. A pooled analysis of population-based surveys from 168 countries representing nine regions from around the world, suggests that approximately 28% of adults aged 18 years and older [3] do not meet the physical activity levels recommended by the World Health Organization (WHO). These recommendations are that adults should complete ≥150 minutes of moderate-to-vigorous physical activity, per week [4]. Globally, each year, physical inactivity contributes to 7.2% of total deaths and 69% of these deaths occur in middle-income countries [1]. The economic impact of physical inactivity is substantial, with global costs reaching INT$ 54 billion in direct healthcare expenses and INT$ 14 billion in lost productivity, per year [5].

Thailand, an upper-middle-income country in Southeast Asia, is experiencing increasingly high rates of physical inactivity and sedentary behavior and a growing burden of NCDs [6–10]. The proportion of adults who engage in sufficient physical activity has declined over time. In 2009 and 2015, 81.5% [8] and 80.8% [6] of the population were classified as sufficiently active, respectively. To address this, in 2018 the Thai government launched the National Physical Activity Strategy 2018–2030 [11] alongside various physical activity-promoting initiatives in cities across the country [12]. However in 2020, and potentially due to restrictions associated with the COVID-19 pandemic [13], the proportion of adults who engaged in sufficient

physical activity declined to 69.1% in 2020 [7] and then increased slightly to 71.9% in 2021 [14]. Rates of sedentary behavior are also high, a national survey conducted in 2021 indicated that 75.8% of adults in Thailand were classified as highly sedentary (defined as being sedentary for $\geq$7 hours per day) [14].

Bangkok is the capital and most populous city of Thailand. Rapid and significant urbanization of the city has potentially had negative implications for physical activity, sedentary behavior [15], and the prevalence of NCDs [16]. Bangkok is one of the fastest growing urban centers in Southeast Asia. In 1950, the city was inhabited by just 1.4 million people [17]. As of 2023, the population has surpassed 11 million people [17], which accounts for approximately 16% of the country's population [18]. In Thailand, existing epidemiological investigations of physical activity and sedentary behavior have predominantly been conducted at the national level [19–22]. Limited studies have focused specifically on Bangkok, and those that are available have concentrated on specific aspects of physical activity, such as exercise or transport behavior, rather than total physical activity and sedentary behavior [23–27]. Bangkok's urbanization is part of a global trend. Understanding how urbanization influences physical activity and sedentary behavior in Bangkok can provide insights into similar trends in other urban centers worldwide, and will be particularly relevant for cities in other countries that are experiencing rapid urbanization.

Physical inactivity and sedentary behavior are independent risk factors for NCDs and premature mortality [2, 28]. However, when these risk factors coexist, they may have a synergistic effect that exacerbates their impact [29]. Consequently, it is important to identify patterns of combined physical activity and sedentary behaviors. Such findings can help inform and improve public health strategies and policies in urban areas.

Therefore, this study aims to investigate the prevalence of combined physical activity and sedentary behavior patterns and the associations between these behavior patterns and socio-demographic characteristics, of residents of Bangkok.

## Method

### Sample and procedure

This study uses data from the nationally representative 2021 Thai Health Behavior Survey, which was conducted by the National Statistical Office (NSO) to evaluate the prevalence of NCDs and associated risk factors (tobacco use, alcohol consumption, unhealthy diet consumption, physical activity, and sedentary behavior) in the Thai population [30]. Random sampling was used to identify households to be invited to participate in a computer-assisted personal interview. Recruitment was stratified to ensure national coverage and representation at the provincial level (covering all 77 provinces, including Bangkok). Participants were eligible for the interview if they were present in the household and were aged 6 years old or above. Questions related to physical activity and sedentary behavior were only collected from participants aged 15 years old or above. Each interview lasted between 60–90 minutes. All interviews were conducted between March and May 2021.

For the current study, we included data from participants aged between 18 and 80 years to align with the age range for the WHO's physical activity and sedentary behavior guidelines for adults [4], and because there were concerns related to the accuracy of data provided by the very elderly [31].

### Measures

**Participants' characteristics.** Participants provided information on their sex (male, female), age (in years), marital status, education, occupation, and monthly income. Age was

classified into three categories (18–45, 46–59, 60–80 years old), marital status was classified into two categories (single/divorced/separated/widowed, married/cohabiting), education level was classified into two categories (<secondary education, ≥secondary education), and occupation was classified into three categories (unemployed/retired, office-based, labor-intensive). Monthly individual income was dichotomized (< 12,000, ≥ 12,000), based on the median (median = 12,000 baht, US$ 1 = 35 baht).

Health status information included body mass index (BMI) and chronic physical condition (s). BMI was calculated from the respondents' self-reported weight and height, and then categorized into (i) healthy weight (BMI <23 kg/m$^2$) or (ii) overweight and obese (BMI > = 23 kg/m$^2$), following BMI classifications for Asian populations [32]. For chronic physical conditions, respondents indicated whether a physician had diagnosed them with any of the following nine chronic conditions: hypertension, diabetes mellitus, hyperlipidemia, myocardial infarction, stroke, chronic obstructive pulmonary disease, cancer, depression, or osteoarthritis. For each respondent, the total number of chronic conditions was calculated, and individuals were then categorized as (i) having no chronic condition, (ii) having one chronic condition, or (iii) having multimorbidity (i.e., two or more chronic conditions).

**Physical activity and sedentary behavior.** The Global Physical Activity Questionnaire (GPAQ) [33] was used to assess levels of physical activity and sedentary behavior. Respondents reported the frequency and duration of moderate and vigorous-intensity physical activity engaged in as part of their work, or for transport or recreation during a typical week. The WHO physical activity guidelines were used to classify whether participants were sufficiently active or not [4]. 'Sufficiently active' was classified as completing at least 150 minutes of moderate-intensity physical activity or 75 minutes of vigorous-intensity physical activity, or an equivalent combination of both, throughout the week.

The GPAQ includes one question on sedentary behavior: "How much time do you usually spend sitting or reclining on a typical day?". Respondents who self-reported ≤7 hours per day of sedentary behavior were categorized as 'low sedentary behavior', based on recent evidence that this threshold was associated with a lower risk of mortality [2].

The GPAQ has demonstrated acceptable convergent validity (Spearman's rho = 0.33, p-value <0.01) when compared with accelerometer-based measurements of physical activity, and has good seven-day test-retest reliability (Spearman's rho = 0.77, p-value <0.01) amongst adults in Thailand [34].

**Data management and statistical analysis.** GPAQ data were analyzed in accordance with the analysis guide [33]. Invalid data, such as instances where respondents reported being active for more than 7 days per week or for more than 16 hours per day, were excluded from analysis. Respondents with inconsistent or missing physical activity data or sociodemographic data were also excluded from the analysis.

Descriptive analysis was conducted to assess the frequency and percentage of participants across four mutually exclusive movement patterns of physical activity and sedentary behavior combinations: (i) highly active/low sedentary (participants with sufficient physical activity and low levels of sedentary behavior), (ii) highly active/highly sedentary (participants with sufficient physical activity and high levels of sedentary behavior), (iii) low active/low sedentary (participants with insufficient physical activity and low levels of sedentary behavior), and (iv) low active/highly sedentary (participants with insufficient activity and high levels of sedentary behavior). The association between participants' characteristics and movement patterns was examined using the chi-square test, and the resulting statistical significance was reported as a p-value.

Multinomial logistic regression was used to examine relationships between participants' characteristics and the four movement patterns of physical activity and sedentary behavior

combinations, with 'low active/highly sedentary' as the reference group. All analyses employed survey weights to account for respondents in each household, non-response, and post-stratification, consistent with the survey methods [30]. Associates were reported as adjusted odds ratio (AOR), with 95% confidence intervals (95% CI), and p-values.

Analyses were conducted in Stata Statistical Software version 17 (StataCorp LP, College Station, TX, USA).

### Ethics approval and consent

In accordance with the Statistics Act, B.E.2550 (2007) [35], which mandated NSO to conduct regular population surveys, ethical review and approval were not required. Respondent consent was also waived, as the survey was conducted by the NSO as part of its institutional and legal mandate [8]. The research team was authorized by the NSO to access the survey microdata for the purpose of conducting this research. Additionally, the Institutional Review Board of the Department of Health, Ministry of Public Health Thailand granted this study a research ethics exemption (No. 533/2565) in July 2022.

## Results

### Study participants

There were 8,538 Bangkok residents who were screened for eligibility for this study. Potential participants were excluded based on age (either below 18 years or above 80 years of age, n = 1,892), for not being present at home on the interview dates (n = 3,345), for providing invalid data (n = 14) or incomplete sociodemographic data (n = 150). The final sample size for analysis consisted of 3,137 participants (Fig 1).

### Participant characteristics

The characteristics of participants are shown in Table 1. The majority were female (55.2%), aged 18–45 years (52.2%), married (54.3%), had completed secondary education (65.4%), were employed in office-based work (46.5%), and were free of chronic medical conditions (74.0%). Based on BMI, roughly equal proportions of participants were classified as having a healthy weight or having overweight/obesity.

### Overall proportion of combined physical activity and sedentary behavior patterns

The majority of participants were categorized as being highly active/highly sedentary (64.8%), followed by being highly active/low sedentary (17.9%) and then being low active/highly sedentary (14.3%) (Table 1). Only a few participants (3.0%) were categorized as low active/low sedentary.

### Proportion of combined physical activity and sedentary behavior patterns by participants' characteristics

Participant characteristics varied across the four movement patterns (Fig 2). The highest proportion of participants being highly active/low sedentary were participants with labor-intensive employment (22.5%), while the lowest proportions were observed among unemployed/retired participants (14.7%) and participants with multimorbidity (14.7%). More females (18.6%) belonged to this category as compared to males (17.1%), and participants with lower levels of education (21.0%) had a higher proportion of being in this category than those with

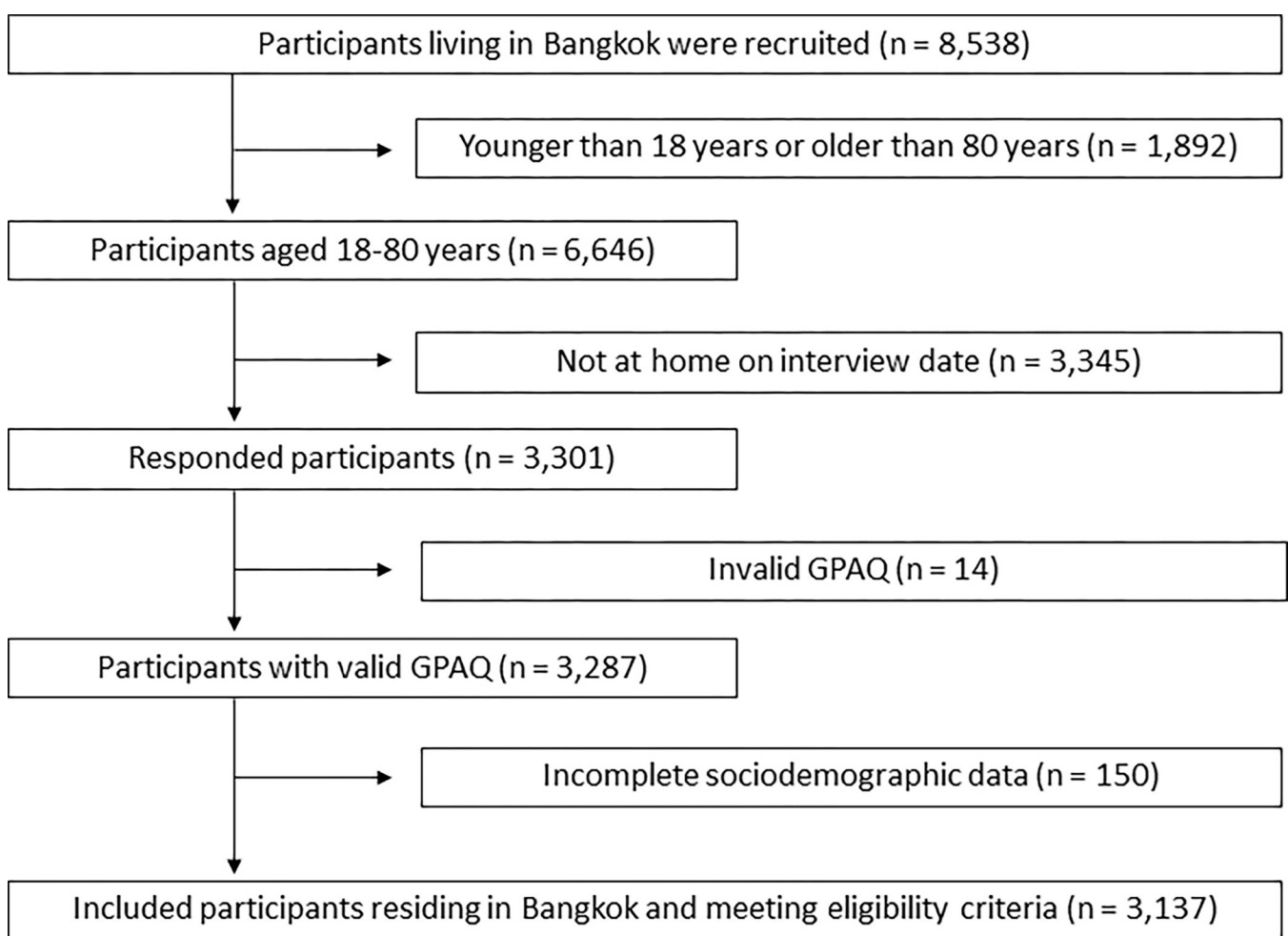

**Fig 1. Study participant flow chart.**

higher education (16.3%). For the highly active/highly sedentary group, the highest proportion was observed among those with higher levels of education (67.0%), while the lowest was among those with lower levels of education (60.6%). For being low active/low sedentary, the highest proportion was observed among participants aged 60–80 years (4.4%) and those with multimorbidity (4.4%), equally. TheT lowest proportion was in participants aged 18–45 years (2.4%). Lastly, for being low active/highly sedentary, the highest proportion was among participants with multimorbidity (17.8%), followed by male participants (16.7%) and participants aged 60–80 years (15.9%), respectively. While the lowest proportion was found in female participants (12.4%), followed by labor-intensive participants (12.9%) and participants aged 18–45 years (13.1%), respectively.

## Association between combined physical activity and sedentary behavior patterns and participants' characteristics: Multinomial logistic regression

Results from the multinomial logistic regression, examining the association between the four groups of physical activity and sedentary behavior combinations and participants' characteristics, were presented in Table 2. Compared to males, females had a significantly higher likelihood of belonging to either the highly active/low sedentary group (AOR = 1.69, 95%CI: 1.25,

**Table 1. Participant characteristics, all participants and according to the four physical activity and sedentary behavior combination movement patterns.**

| Participants' characteristics | Overall | | Highly active/low sedentary | | Highly active/highly sedentary | | Low active/low sedentary | | Low active/highly sedentary | | p-value[a] |
|---|---|---|---|---|---|---|---|---|---|---|---|
| | N = 3,137 (100%) | | n = 562 (17.9%) | | n = 2,032 (64.8%) | | n = 94 (3.0%) | | n = 449 (14.3%) | | |
| | n | Weighted % | n | Weighted % | n | Weighted % | n | Weighted % | n | Weighted % | |
| Overall sample | 3,137 | 100.0 | 562 | 100.0 | 2,032 | 100.0 | 94 | 100.0 | 449 | 100.0 | |
| **Sex** | | | | | | | | | | | **0.001** |
| Male | 1,404 | 44.8 | 239 | 42.6 | 880 | 43.3 | 50 | 53.1 | 235 | 52.2 | |
| Female | 1,733 | 55.2 | 323 | 57.4 | 1,152 | 56.7 | 44 | 46.9 | 214 | 47.8 | |
| **Age** (years) | | | | | | | | | | | 0.17 |
| 18–45 | 1,639 | 52.2 | 308 | 54.7 | 1,076 | 53.0 | 40 | 42.4 | 215 | 48.0 | |
| 46–60 | 888 | 28.3 | 155 | 27.6 | 569 | 28.0 | 27 | 28.9 | 137 | 30.4 | |
| 61–80 | 610 | 19.5 | 99 | 17.7 | 387 | 19.0 | 27 | 28.7 | 97 | 21.6 | |
| **Marital status** | | | | | | | | | | | 0.54 |
| Single/divorced/separated/widowed | 1,432 | 45.7 | 254 | 45.3 | 919 | 45.3 | 42 | 44.7 | 216 | 48.2 | |
| Married/cohabiting | 1,705 | 54.3 | 308 | 54.7 | 1,113 | 54.7 | 52 | 55.3 | 233 | 51.8 | |
| **Education** | | | | | | | | | | | **0.002** |
| Below secondary education | 1,085 | 34.6 | 228 | 40.6 | 657 | 32.3 | 36 | 38.3 | 163 | 36.4 | |
| At least secondarys education | 2,052 | 65.4 | 334 | 59.4 | 1,375 | 67.7 | 58 | 61.7 | 286 | 63.6 | |
| **Occupation** | | | | | | | | | | | **0.003** |
| Unemployed/retired | 811 | 25.9 | 119 | 21.2 | 543 | 26.7 | 28 | 30.2 | 121 | 26.9 | |
| Office-based workers | 1,459 | 46.5 | 248 | 44.0 | 955 | 47.0 | 40 | 42.2 | 217 | 48.3 | |
| Labor-intensive workers | 867 | 27.6 | 195 | 34.8 | 534 | 26.3 | 26 | 27.6 | 111 | 24.8 | |
| **Income** (median: baht/month) | | | | | | | | | | | 0.90 |
| <12,000 | 1,457 | 46.5 | 262 | 46.5 | 959 | 47.2 | 38 | 40.0 | 199 | 44.4 | |
| > = 12,000 | 1,680 | 53.5 | 300 | 53.5 | 1,073 | 52.8 | 56 | 60.0 | 250 | 55.6 | |
| **Body mass index** (BMI) | | | | | | | | | | | 0.49 |
| Healthy weight (BMI <23) | 1,551 | 49.5 | 273 | 48.6 | 994 | 48.9 | 53 | 56.9 | 231 | 51.5 | |
| Overweight/obesityo (BMI > = 23) | 1,586 | 50.5 | 289 | 51.4 | 1,038 | 51.1 | 41 | 43.1 | 218 | 48.5 | |
| **Chronic physical condition** | | | | | | | | | | | 0.10 |
| No chronic condition | 2,320 | 74.0 | 435 | 77.3 | 1,504 | 74.0 | 64 | 68.6 | 318 | 70.7 | |
| One chronic condition | 440 | 14.0 | 72 | 12.8 | 290 | 14.3 | 13 | 13.5 | 64 | 14.4 | |
| Multimorbidity | 377 | 12.0 | 55 | 9.9 | 238 | 11.7 | 17 | 17.9 | 67 | 14.9 | |

[a] Bivariate association between categorical variables and physical activity/ sedentary behavior were examined via chi-square analyses.

2.28) or highly active/highly sedentary group (AOR = 1.51, 95%CI: 1.19, 1.93) as compared to the low active/high sedentary group. Labor-intensive participants, when compared to unemployed/retired participants, had a significantly higher likelihood of being in the highly active/low sedentary group rather than the low active/highly sedentary group (AOR = 1.89, 95%CI: 1.22, 2.94). Participants who reported multimorbidity had a significantly lower likelihood of being in the highly active/low sedentary group compared to the low active/highly sedentary (AOR = 0.60, 95%CI: 0.37, 0.98), as compared to those with no chronic physical conditions.

## Discussion

This study investigated patterns and correlates of physical activity and sedentary behavior among the adult population in Bangkok using the most up-to-date nationally representative data. Result indicate that the largest proportion of the study participants belonged to the highly active/highly sedentary group, followed by the highly active/low sedentary group. The study

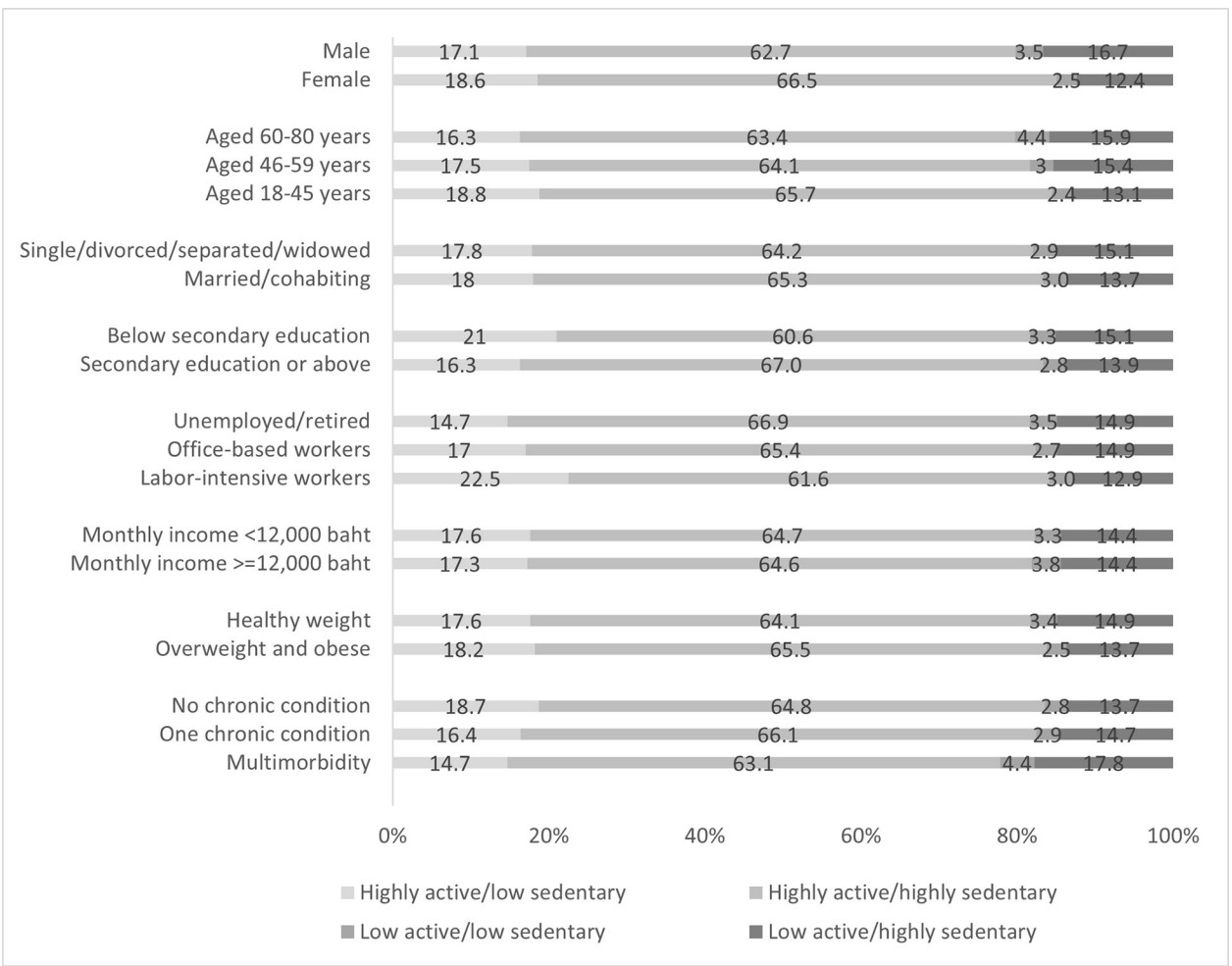

**Fig 2. The proportion of four combined physical activity and sedentary behavior patterns by participants' characteristics.**

also identified associations between sex, occupation type, and presence of chronic physical conditions, and their influence on the likelihood of an individual belonging to each of the four combined physical activity and sedentary behavior patterns.

The findings suggest that less than one-fifth (17.9%) of Bangkok residents may achieve the recommended levels of physical activity engage in low levels of sedentary behavior, the combination with the greatest benefit for health [29]. However, two-thirds of participants reported sufficient physical activity but still engaged in high levels of sedentary behavior. This presents a significant public health concern since sedentary behavior, regardless of physical activity levels, is a risk factor for NCDs and increases the risk of all-cause mortality [2, 36]. Therefore, it would be beneficial to prioritize efforts towards mobilizing the large proportion of the population who are currently classified as being highly active/highly sedentary or low active/highly sedentary, to reduce their sedentary time. This strategy is aligned with the WHO concept of 'every move counts' [4] and could serve as an initial health promotion phase, which would be followed by targeting physical activity during subsequent phases. Interventions to replace sedentary behavior with light-to-moderate-intensity movement several times a day could be developed. Examples include using screen time control measures such as electronic lock-out systems on televisions, computers, or smartphones [37]. Furthermore, promoting urban

**Table 2. Multinomial logistic regression analyses of association between movement patterns of physical active and sedentary behavior combinations and participants' correlates (weighted).**

| Correlates | Highly active/low sedentary (n = 562) | | | | Highly active/highly sedentary (n = 2,032) | | | | Low active/low sedentary (n = 94) | | | |
|---|---|---|---|---|---|---|---|---|---|---|---|---|
| | AOR | 95% CI | | p | AOR | 95% CI | | p | AOR | 95% CI | | p |
| | | Lower | Upper | | | Lower | Upper | | | Lower | Upper | |
| **Sex** | | | | | | | | | | | | |
| female | **1.69** | 1.25 | 2.28 | **0.001** | **1.51** | 1.19 | 1.93 | **0.001** | 1.02 | 0.66 | 1.60 | 0.92 |
| ref = male | | | | | | | | | | | | |
| **Age** (years) | | | | 0.71 | | | | 0.41 | | | | 0.11 |
| 46–59 | 0.80 | 0.57 | 1.13 | 0.20 | 0.84 | 0.64 | 1.11 | 0.22 | 1.11 | 0.63 | 1.98 | 0.72 |
| 60–80 | 0.97 | 0.63 | 1.49 | 0.90 | 0.88 | 0.62 | 1.24 | 0.46 | 1.65 | 0.88 | 3.06 | 0.12 |
| ref = 18–45 | | | | | | | | | | | | |
| **Marital status** | | | | | | | | | | | | |
| married/co-habiting | 1.10 | 0.83 | 1.46 | 0.51 | 1.15 | 0.91 | 1.45 | 0.24 | 1.14 | 0.72 | 1.80 | 0.58 |
| ref = single/divorced/separated/widowed | | | | | | | | | | | | |
| **Education** | | | | | | | | | | | | |
| at least secondary education | 0.90 | 0.66 | 1.24 | 0.53 | 1.27 | 0.98 | 1.63 | 0.07 | 0.94 | 0.54 | 1.62 | 0.82 |
| ref = below secondary education | | | | | | | | | | | | |
| **Occupation** | | | | **0.005** | | | | 0.35 | | | | 0.87 |
| office-based workers | 1.19 | 0.80 | 1.77 | 0.39 | 1.00 | 0.74 | 1.36 | 0.99 | 0.77 | 0.44 | 1.34 | 0.35 |
| labor-intensive workers | **1.89** | 1.22 | 2.94 | **0.005** | 1.20 | 0.83 | 1.73 | 0.33 | 1.03 | 0.54 | 1.95 | 0.93 |
| ref = unemployed/retired | | | | | | | | | | | | |
| **Income** | | | | | | | | | | | | |
| > = 12,000 baht/month | 0.92 | 0.66 | 1.28 | 0.63 | 0.86 | 0.67 | 1.11 | 0.25 | 1.52 | 0.92 | 2.51 | 0.10 |
| ref = <12,000 baht/month | | | | | | | | | | | | |
| **Body mass index** (BMI) | | | | | | | | | | | | |
| BMI > = 23 | 1.24 | 0.93 | 1.66 | 0.14 | 1.20 | 0.94 | 1.52 | 0.14 | 0.77 | 0.49 | 1.21 | 0.26 |
| ref = BMI <23 | | | | | | | | | | | | |
| **Chronic physical condition** | | | | 0.06 | | | | 0.31 | | | | 0.93 |
| one chronic condition | 0.84 | 0.55 | 1.27 | 0.41 | 0.99 | 0.71 | 1.40 | 0.97 | 0.83 | 0.42 | 1.62 | 0.58 |
| multimorbidity | **0.60** | 0.37 | 0.98 | **0.04** | 0.79 | 0.54 | 1.14 | 0.21 | 1.03 | 0.54 | 1.95 | 0.93 |
| ref = no chronic condition | | | | | | | | | | | | |

AOR = adjusted odds ratio, 95% CI = 95% confidence interval, p = p-value

Ref = low active/highly sedentary (n = 449)

planning strategies that prioritize walkable cities, enhance public transport systems, and establish neighborhoods and urban areas that integrate residential, commercial, and recreational spaces, are also advisable [38–41].

Our results indicate that females had a higher likelihood of being either highly active/low sedentary or highly active/highly sedentary, as compared to males. This contrasted with the results of a national survey conducted in Thailand in 2015, which indicated that males were more likely to achieve sufficient physical activity [19]. Global trends from 2001–2016 also suggested that the prevalence of sufficient physical activity was higher in men (76.6%) than in women (68.3%) [3]. However, the discrepancy between the previous and current reporting of physical activity in women could be due to changes in the workforce. Employment of women in the formal industrial section in Bangkok [42] has increased by 0.8 million females, representing around 12% growth over the past seven years. The majority of these women are working in the manufacturing, or in wholesale and retail trade (26.4% and 16.7%, respectively) [43].

This trend aligns with other upper-middle-income and high-income countries, where the majority of women (64% and 67%, respectively) are now participating in workforce [44]. This shift in women's employment provides opportunities for women to leave their homes for workplaces and may increase their physical movements, particularly for transport purposes [10]. It should also be acknowledged that data collection for this survey was conducted during the implementation of social distancing measures in response to the COVID-19 pandemic. These measures influenced the physical mobility of individuals', where trips decreased by 11% (from 9,580 million in 2020 to 8,522 million in 2021) [45]. This could result in a decrease in physical activity and an increase in sedentary behavior [13, 46].

Furthermore, this study indicates that labor-intensive occupations had a higher likelihood of being highly active/low sedentary compared to other occupations. This finding is consistent with prior research on occupational physical activity conducted globally [47, 48], in Asia [49–51], and specifically in Thailand [20, 21]. This phenomenon can be attributed to the physical demands of labor-intensive work. In contrast, unemployed or retired individuals, as well as office-based employees, often experience less physical demand and instead engage in prolonged periods of sitting. Notably, the rapid progress of urbanization, particularly in low- and middle-income countries [15], has led to a shift in the labor sector away from labor-intensive jobs and towards more sedentary occupations [48, 52, 53]. The proportion of labor-intensive occupations dropped substantially from 2015 to 2021 (from 23.8 to 19.5, respectively). Similarly, the proportion of office-based occupations increased substantially during the same period (from 14.2 to 18.0 million) [43]. Consequently, policy interventions should focus on enhancing opportunities and improving environmental factors that facilitate physical activity while minimizing sedentary behavior in the workplace [54]. Examples include using sit-stand desks, treadmill desks, or cycling desks combined with educational information, counseling, and short breaks or walking strategies [55]. Implementing point-of-choice prompting software along with educational initiatives may also prove effective [55]. Social-level components such as team movement breaks with incentives, like lottery rewards in Thailand can further encourage behavioral change [56].

Participants with multimorbidity had a significantly lower likelihood of being highly active/low sedentary as compared to those without chronic physical conditions. This finding was consistent with the previous Thai national survey in 2015 [19] and a study conducted on the multiethnic Asian population in Singapore [50]. It may be that the presence of multiple physical limitations, pain, and fatigue that are often associated with multimorbidity reduces time spent active and increases time spent sedentary [57, 58]. Additionally, the cumulative burden of multiple chronic conditions can have a psychological impact on individuals, leading to increased stress, anxiety, and depression, and thereby exacerbating the challenges faced in engaging in regular physical activity [59]. Effectively addressing these barriers necessitates the implementation of customized exercise programs and comprehensive support systems [59, 60].

The strengths and limitations of this study should be acknowledged. A key strength was the utilization of data from the NSO, which provided a large representative sample of households in Bangkok. This approach facilitated robust estimations of the prevalence of physical activity and sedentary behavior. Limitations should also be considered when interpreting the findings. Firstly, the reliance on self-reported data using the GPAQ introduced the potential for memory bias. Participants may have difficulty accurately recalling their physical activity and sedentary behavior over the previous seven days, leading to inaccuracies in the reported prevalence rates. Furthermore, self-report measures were subject to socially desirable biases [61]. Secondly, this study is cross-sectional in nature, thereby precluding the establishment of causal relationships between the independent variables and the outcomes. Lastly, the implementation of COVID-19 physical and social distancing measures during the data collection period may have had an impact on the prevalence of physical activity and increased sedentary behavior.

## Conclusion

This study examines the patterns and factors associated with physical activity and sedentary behavior among the population in Bangkok, using up-to-date nationally representative data. The findings indicate that the majority of participants belong to the highly active/highly sedentary group, followed by the highly active/low sedentary group. Several correlates, including sex, occupation, and chronic physical conditions, were found to be associated with different physical activity and sedentary behavior patterns. These findings highlight the need for targeted interventions to create supportive environments that facilitate physical activity and reduce sedentary behavior in recreational activities, workplaces, and urban settings, such as implementing screen time control measures and movement breaks. Additionally, enhancing the built environment in Bangkok, including promoting walkable cities and improving public transport systems, can play a significant role in addressing these issues.

## Acknowledgments

The authors would like to express their gratitude to the National Statistical Office for providing the valuable database and to all participants and staff involved in the study. We reserve special thanks to Dr Nicholas Alexander Petrunoff, Dr Borame Sue Lee Dickens from the National University of Singapore, Ms Orana Chandrasiri, Dr Sigit Arifwidodo from the Kasetsart University, and Dr Vuthiphan Vongmongkol and Ms Jintana Jankhotkaew from the International Health Policy Program for their unrelenting support and invaluable advice throughout the study course.

## Author Contributions

**Conceptualization:** Thitikorn Topothai, Viroj Tangcharoensathien, Sarah Martine Edney, Rapeepong Suphanchaimat, Orratai Waleewong, Chompoonut Topothai, Falk Müller-Riemenschneider.

**Data curation:** Thitikorn Topothai.

**Formal analysis:** Thitikorn Topothai, Rapeepong Suphanchaimat.

**Funding acquisition:** Thitikorn Topothai.

**Investigation:** Thitikorn Topothai.

**Methodology:** Thitikorn Topothai, Rapeepong Suphanchaimat.

**Project administration:** Thitikorn Topothai, Orratai Waleewong.

**Resources:** Thitikorn Topothai.

**Software:** Thitikorn Topothai.

**Supervision:** Viroj Tangcharoensathien, Falk Müller-Riemenschneider.

**Validation:** Thitikorn Topothai.

**Visualization:** Thitikorn Topothai.

**Writing – original draft:** Thitikorn Topothai.

**Writing – review & editing:** Thitikorn Topothai, Viroj Tangcharoensathien, Sarah Martine Edney, Rapeepong Suphanchaimat, Angkana Lekagul, Orratai Waleewong, Chompoonut Topothai, Anond Kulthanmanusorn, Falk Müller-Riemenschneider.

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
