## [Decision Letter · Decision Letter 0]

23 Aug 2023

PONE-D-23-18854Patterns and Correlates of Physical Activity and Sedentary Behavior Among Bangkok Residents: A Cross-sectional StudyPLOS ONE

Dear Dr. **Topothai**,

Thank you for submitting your manuscript to PLOS ONE. After careful consideration, we feel that it has merit but does not fully meet PLOS ONE’s publication criteria as it currently stands. Therefore, we invite you to submit a revised version of the manuscript that addresses the points raised during the review process.

ACADEMIC EDITOR:1. In the background section the third paragraph on demographics of Bangkok could be shortened to two to three sentences.2. In the discussion section you can add what types of jobs are females mostly employed in Thailand, also you can tie in that with lifestyles of females in Thailand (like Proportion of single females, single parents, office workers and educational status).  Also could you discuss what proportion of thai population are office workers or employed in labour intensive jobs, this might have implications for national physical activity policy.3. You have recommended the environmental restructuring interventions like sit-stand desk which have been shown to be effective only in short term in various studies and completely ignored the incidental physical activity, like taking stairs instead of escalators, placing bins and printers further away from desk. Please refer to the publication below for interventions for reducing sedentary behaviour and also please add what can be done to reduce sedentary behaviour in leisure time.https://www.cochranelibrary.com/cdsr/doi/10.1002/14651858.CD010912.pub5/fullhttps://bjsm.bmj.com/content/53/19/1206/4. The manuscript should be prof read by proficient English speaker.

We look forward to receiving your revised manuscript.

Kind regards,

Nipun Shrestha, Ph.D.

Academic Editor

PLOS ONE

Journal Requirements:

Reviewers' comments:

Reviewer's Responses to Questions

**Comments to the Author**

1. Is the manuscript technically sound, and do the data support the conclusions?

Reviewer #1: Yes

Reviewer #2: Yes

2. Has the statistical analysis been performed appropriately and rigorously? 

Reviewer #1: Yes

Reviewer #2: Yes

3. Have the authors made all data underlying the findings in their manuscript fully available?

Reviewer #1: Yes

Reviewer #2: Yes

4. Is the manuscript presented in an intelligible fashion and written in standard English?

Reviewer #1: Yes

Reviewer #2: No

5. Review Comments to the Author

Reviewer #1: Dear editors, thank you for inviting me reviewing this interesting manuscript. My suggestions are: Minor revision needed. Below are my comments:

• Comments 1:

Page 3, Background:

“The prevalence of inadequate physical activity levels is

a matter of concern, affecting approximately 28% of adults (equivalent to 1.4 billion individuals)(3) failing to meet the recommended physical activity levels outlined by the World Health Organization (WHO) (defined as ≥150 minutes of moderate-to-vigorous physical activity per week).(4)”

Could you provide clarity on where this survey was conducted, or is this research about global trends?

• Comments 2:

Page 3, Background:

“Furthermore, in 2020 on average Thai individuals spent approximately 14 hours sedentary daily, (9)”

This sentence seems unclear for me, please reconstruct it.

• Comments 3:

Page 3, Background:

“In contrast, Chon Buri, Thailand’s second most populated city, has a population of just 1.5 million. (16)”

This sentence does not add different and useful information and it is recommended to delete it.

• Comments 4:

Page 4, Method, Sample and procedure:

The authors say that this research is about residents of Bangkok. However, it seems difficult to identify which cities were investigated in the 2021 Thai Health Behavior Survey in the Methods section. Please clarify this for now.

• Comments 5:

Page 4, Method, Sample and procedure:

“For the current study, we include data from participants aged between 18 to 80 years to enable comparison with global recommendations on physical activity and sedentary behavior levels for adults.”

Please add a reference to support your statement.

• Comments 6:

Page 5, Method, Physical Activity and Sedentary Behavior:

“The GPAQ has undergone validation in the adult population of Thailand, demonstrating an acceptable criterion validity (Spearman's rho = 0.33, p-value <0.01) with accelerometer-based measurements of physical activity.”

Please read this paper: https://pubmed.ncbi.nlm.nih.gov/26931142/.

Validation is not an appropriate word to describe the relationship between these two measures. It may be better to use “convergent validity” or ‘agreement between measures”.

• Comments 7:

Page 5, Method, Physical Activity and Sedentary Behavior:

“Additionally, validation studies conducted among a multi-ethnic population in

Singapore, indicating a moderately correlated (Spearman's rho = 0.39, p-value <0.001) of moderate-tovigorous physical activity and sedentary behavior (Spearman's rho = 0.28, p-value <0.05) with accelerometer-based measurements..”

This sentence will confuse readers who do not have sufficient geographical knowledge. Please elaborate further on the relationship between Thailand and Singapore, e.g. geography, culture, ethnic composition, etc.

• Comments 8:

Page 6, Results, Study Participants:

“Of these, some were excluded based on age (either below 18 years or above 80 years of age, n=1892), for not being present at their homes on the interview dates (n=3,345), or for proving invalid data (n=14), or missing data (n=150).”

Of these is too colloquial and could be used e.g. "according to the inclusion and exclusion criteria". It is suggested that this sentence be reorganised.

• Comments 9:

Page 15, Figure 1:

Missing unit of measurement in the age column; BMI lacks full spelling; The proportion of the sample in the low active/low sedentary group should be 3.0%, keeping one decimal place as in the other groups.

• Comments 10:

Page Figure 2:

The information within this figure is an exact duplicate of that in Table 1 and is recommended to be deleted.

• Comments 11:

Page 7, Discussion:

“which was the most health-enhancing combination of sufficient physical activity and low sedentary behavior”

Please provide evidence to support this statement.

• Comments 12:

Page 7, Discussion:

“It would be helpful to prioritize efforts towards mobilizing the large proportion of the population who are currently classified as being highly active/highly sedentary, to be less sedentary.”

Why this population? Are there studies that compare the risks of different combinations and health-related outcomes? Do group with low physical activity and high sedentary behaviour also need to be prioritised for intervention?

• Comments 13:

Page 8, Discussion:

“This finding was consistent with the previous Thai national survey in 2015

(25) and a study conducted on the multi-ethnic Asian population in Singapore,

(47).”

Attention needs to be paid to in-text citations and punctuation, and it is recommended that the text be critically edited.

• Comments 14:

Page 9, Discussion:

“Firstly, the reliance on self-reported data using the GPAQ introduced the potential for memory bias. Participants may have difficulty accurately recalling their physical activity and sedentary behavior over the previous seven days, leading to inaccuracies in the reported prevalence rates.”

This is indeed a limitation of self-reported measurements, but are accelerometer measurements subject to recall bias?

Reviewer #2: This is an interesting cross-sectional study determining patterns and correlates of physical activity and sedentary behaviour among individuals in Bangkok. The study was well-written and coherent. However, I suggest following recommendations in order to improve the manuscript.

Introduction:

While there have been nationally representative studies from Thailand conducted previously on the topic, the need for this study specifically focusing Bangkok is not yet clear. Although authors have attempted to explain it, the reason why it is important to study PA and SB in individuals from Bangkok is important should be explained better.

Methods:

Page 4: What was the rationale for categorising individual income based on 12000 Baht? Please provide a reference.

Results:

Page 6: The numbers don’t add to 8,538. Please check and correct.

Discussion:

Page 7: While I agree that financial incentives have the potential to influence individual behaviour, it might be argued that it is not a cost-effective approach. Could you suggest better alternatives such as changes to the workstations (e.g., sit-to-stand desks) that are cost-efficient? Use of stairs instead of escalators?

Page 8: Why could covid-19 changes have impacted PA and SB in men compared to women?

Tables:

Table 2 need correction: please remove the bullets and format the text in sentence case.

English and grammar:

The use of English was poor at several places and should be significantly improved. I suggest the manuscript be reviewed by a proficient English speaker.

6. PLOS authors have the option to publish the peer review history of their article (what does this mean?). If published, this will include your full peer review and any attached files.

Reviewer #1: No

Reviewer #2: No

---

## [Author Response · Author response to Decision Letter 0]

11 Sep 2023

Editor: In the background section the third paragraph on demographics of Bangkok could be shortened to two to three sentences. 

Thanks for the comment. We’ve shortened the third paragraph and merged it with the fourth paragraph (page 4).

“Bangkok is the capital and most populous city of Thailand. Rapid and significant urbanization of the city has potentially had negative implications for physical activity, sedentary behavior [15], and the prevalence of NCDs [16]. Bangkok is one of the fastest growing urban centers in Southeast Asia. In 1950, the city was inhabited by just 1.4 million people [17]. As of 2023, the population has surpassed 11 million people [17], which accounts for approximately 16% of the country’s population [18]. In Thailand, existing epidemiological investigations of physical activity and sedentary behavior have predominantly been conducted at the national level [19-22]. Limited studies have focused specifically on Bangkok, and those that are available have concentrated on specific aspects of physical activity, such as exercise or transport behavior, rather than total physical activity and sedentary behavior [23-27]. Bangkok's urbanization is part of a global trend. Understanding how urbanization influences physical activity and sedentary behavior in Bangkok can provide insights into similar trends in other urban centers worldwide, and will be particularly relevant for cities in other countries that are experiencing rapid urbanization.”

Editor: In the discussion section you can add what types of jobs are females mostly employed in Thailand, also you can tie in that with lifestyles of females in Thailand (like Proportion of single females, single parents, office workers and educational status). 

We’ve provided statistics on female occupations (page 15).

“Employment of women in the formal industrial section in Bangkok [42] has increased by 0.8 million females, representing around 12% growth over the past seven years. The majority of these women are working in the manufacturing, or in wholesale and retail trade (26.4% and 16.7%, respectively) [43].”

There is a qualitative study on Thai female lifestyle, but it did not demonstrate these proportions.

Editor: Also could you discuss what proportion of thai population are office workers or employed in labour intensive jobs, this might have implications for national physical activity policy. 

We’ve provided information on the proportion of office-based and labor-intensive occupations (page 16).

“The proportion of labor-intensive occupations dropped substantially from 2015 to 2021 (from 23.8 to 19.5, respectively). Similarly, the proportion of office-based occupations increased substantially during the same period (from 14.2 to 18.0 million) [43].

Editor: You have recommended the environmental restructuring interventions like sit-stand desk which have been shown to be effective only in short term in various studies and completely ignored the incidental physical activity, like taking stairs instead of escalators, placing bins and printers further away from desk. Please refer to the publication below for interventions for reducing sedentary behaviour and also please add what can be done to reduce sedentary behaviour in leisure time. 

We’ve provided examples of sedentary-breaking interventions according to your suggested references in the 2nd paragraph of the discussion (pages 15-16).

“…Interventions could be developed to replace sedentary behavior with light-to-moderate-intensity movement several times a day. Examples include using screen time control measures such as electronic lock-out systems to television, computer or smartphone [34]…”

We’ve provided examples of sedentary-breaking interventions according to your suggested references in the 4th paragraph of the discussion (pages 17-18).

“…Examples include using sit-stand desks, treadmill desks, or cycling desks combined with information, counseling, and short break or walking strategies [52]. Implementing point-of-choice prompting software along with educational initiatives may also prove effective [52]…”

Editor: The manuscript should be prof read by proficient English speaker.

The manuscript has been proofread and revised by a proficient English speaker.

Reviewer #1: Page 3, Background: 

“The prevalence of inadequate physical activity levels is a matter of concern, affecting approximately 28% of adults (equivalent to 1.4 billion individuals)(3) failing to meet the recommended physical activity levels outlined by the World Health Organization (WHO) (defined as ≥150 minutes of moderate-to-vigorous physical activity per week).(4)” 

Could you provide clarity on where this survey was conducted, or is this research about global trends?

Thanks for the comment. We’ve provided information on the setting of the study (page 3).

“…A pooled analysis of population-based surveys in 168 countries from nine regions around the world revealed that approximately 28% of adults aged 18 years and older [3] failed to meet the recommended physical activity levels outlined by...”

Reviewer #1: Page 3, Background: 

“Furthermore, in 2020 on average Thai individuals spent approximately 14 hours sedentary daily, (9)” 

This sentence seems unclear for me, please reconstruct it.

We’ve reconstructed the sentence by cutting the 2020 survey and keeping the 2021 survey (pages 3-4).

“…Furthermore, a national survey conducted in 2021 revealed that 75.8% of Thai individuals were classified as highly sedentary, defined as ≥7 hours per day [14].”

Reviewer #1: Page 3, Background: 

“In contrast, Chon Buri, Thailand’s second most populated city, has a population of just 1.5 million. (16)” 

This sentence does not add different and useful information and it is recommended to delete it.

We’ve deleted the sentence and reconstructed the paragraph as also suggested by the Editor.

Reviewer #1: Page 4, Method, Sample and procedure: 

The authors say that this research is about residents of Bangkok. However, it seems difficult to identify which cities were investigated in the 2021 Thai Health Behavior Survey in the Methods section. Please clarify this for now.

We’ve provided additional information for clarity (page 5).

“…The NSO used random sampling to identify households for participation in computer-assisted personal interviews, which were stratified to ensure national coverage and representation at the provincial level (covering all 77 provinces, including Bangkok)…”

Reviewer #1: Page 4, Method, Sample and procedure: 

“For the current study, we include data from participants aged between 18 to 80 years to enable comparison with global recommendations on physical activity and sedentary behavior levels for adults.” 

Please add a reference to support your statement.

Thanks for the comment. We’ve added references (page 6).

“For the current study, we included data from participants aged between 18 and 80 years, which was consistent with the age range recommended for assessing physical activity and sedentary behavior levels in adults and the elderly [4], and addressed data accuracy concerns among the very elderly [28].”

Reviewer #1: Page 5, Method, Physical Activity and Sedentary Behavior: 

“The GPAQ has undergone validation in the adult population of Thailand, demonstrating an acceptable criterion validity (Spearman's rho = 0.33, p-value <0.01) with accelerometer-based measurements of physical activity.” 

Please read this paper: https://pubmed.ncbi.nlm.nih.gov/26931142/. Validation is not an appropriate word to describe the relationship between these two measures. It may be better to use “convergent validity” or ‘agreement between measures”.

Page 5, Method, Physical Activity and Sedentary Behavior: 

“Additionally, validation studies conducted among a multi-ethnic population in Singapore, indicating a moderately correlated (Spearman's rho = 0.39, p-value <0.001) of moderate-tovigorous physical activity and sedentary behavior (Spearman's rho = 0.28, p-value <0.05) with accelerometer-based measurements..” 

This sentence will confuse readers who do not have sufficient geographical knowledge. Please elaborate further on the relationship between Thailand and Singapore, e.g. geography, culture, ethnic composition, etc.

Thanks for these two comments. We’ve agreed on using ‘convergent validity’. We’ve removed validity and reliability studies in Singapore and kept a study in Thailand. We’ve revised the text for clarity as follows (pages 7-8).

“The GPAQ has demonstrated acceptable convergent validity (Spearman's rho = 0.33, p-value <0.01) with accelerometer-based measurements of physical activity and good seven-day test-retest reliability (Spearman's rho = 0.77, p-value <0.01) in the adult and older adult population of Thailand [31].”

Reviewer #1: Page 6, Results, Study Participants: 

“Of these, some were excluded based on age (either below 18 years or above 80 years of age, n=1892), for not being present at their homes on the interview dates (n=3,345), or for proving invalid data (n=14), or missing data (n=150).” 

Of these is too colloquial and could be used e.g. "according to the inclusion and exclusion criteria". It is suggested that this sentence be reorganised.

Thanks for the comment. We’ve revised the texts as suggested (page 9).

“According to the inclusion and exclusion criteria, some participants were excluded based on age (either below 18 years or above 80 years of age, n=1,892), for not being present at their homes on the interview dates (n=3,345), or for proving invalid data (n=14).”

Page 15, Figure 1: 

Reviewer #1: Missing unit of measurement in the age column; BMI lacks full spelling; The proportion of the sample in the low active/low sedentary group should be 3.0%, keeping one decimal place as in the other groups.

Thanks for the comment. You mean Table 1? We’ve revised Table 1 (and 2) as suggested.

Age (years).

Body mass index (BMI).

Low active/low sedentary, n=94 (3.0%). 

Reviewer #1: Page Figure 2: 

The information within this figure is an exact duplicate of that in Table 1 and is recommended to be deleted.

Thanks for the comment. We’ve removed Fig 2.

Reviewer #1: Page 7, Discussion: 

“which was the most health-enhancing combination of sufficient physical activity and low sedentary behavior” 

Please provide evidence to support this statement.

Thanks for the comment. We’ve removed this clause from paragraph 1 to paragraph 2 of the discussion and provided the reference (page 15).

“The findings suggested that nearly one-fifth (17.9%) of Bangkok residents may achieve the recommended level of physical activity with low sedentary behavior, which represented the most health-enhancing combination [26].”

Bakrania, K., C. L. Edwardson, D. H. Bodicoat, D. W. Esliger, J. M. Gill, A. Kazi, L. Velayudhan, A. J. Sinclair, N. Sattar, S. J. Biddle, et al. "Associations of mutually exclusive categories of physical activity and sedentary time with markers of cardiometabolic health in english adults: A cross-sectional analysis of the health survey for england." BMC Public Health 16 (2016): 25. 10.1186/s12889-016-2694-9. https://www.ncbi.nlm.nih.gov/pubmed/26753523.

Reviewer #1: Comments 12: Page 7, Discussion: 

“It would be helpful to prioritize efforts towards mobilizing the large proportion of the population who are currently classified as being highly active/highly sedentary, to be less sedentary.” 

Why this population? Are there studies that compare the risks of different combinations and health-related outcomes? Do group with low physical activity and high sedentary behaviour also need to be prioritised for intervention?

Thanks for the comment. 

Our approach prioritizes the highly sedentary group before addressing the low physical activity group. The rationale behind this is that while the ease or difficulty of breaking sedentary behavior versus increasing physical activity depends on individual preferences, lifestyles, and starting points, some individuals may find it easier to begin with small changes in sedentary behavior and gradually progress towards increased physical activity, which typically demands more significant physical effort, time commitment, and motivation.

The study by Bakrania et al. provided in the above comment demonstrated the risks of different combinations and health-related outcomes. 

It is true that the group with low physical activity and high sedentary behavior also needs to be prioritized for intervention.

We’ve revised the texts to accommodate comments as follows (page 15).

“…Therefore, it would be beneficial to prioritize efforts towards mobilizing the large proportion of the population who are currently classified as being highly active/highly sedentary or low active/highly sedentary, to reduce their sedentary time, aligning with the WHO concept of ‘every move counts’ [4]. This can serve as an initial step in promoting physical activity during subsequent phases…”

Reviewer #1: Page 8, Discussion: 

“This finding was consistent with the previous Thai national survey in 2015 (25) and a study conducted on the multi-ethnic Asian population in Singapore, (47).” 

Attention needs to be paid to in-text citations and punctuation, and it is recommended that the text be critically edited.

Thanks for the comment. We’ve carefully checked the manuscript for typos.

For this sentence, it has been edited.

“This finding was consistent with the previous Thai national survey in 2015 [19] and a study conducted on the multi-ethnic Asian population in Singapore [47].”

Reviewer #1: Page 9, Discussion: 

“Firstly, the reliance on self-reported data using the GPAQ introduced the potential for memory bias. Participants may have difficulty accurately recalling their physical activity and sedentary behavior over the previous seven days, leading to inaccuracies in the reported prevalence rates.” 

This is indeed a limitation of self-reported measurements, but are accelerometer measurements subject to recall bias?

Thanks for the comment. We understand that accelerometer measurements are not subject to recall bias in the same way that self-reported data can be. Recall bias occurs when individuals have difficulty accurately recalling past events or behaviors from memory, which can lead to inaccuracies in self-reported data. Accelerometer measurements, on the other hand, do not rely on memory or self-reporting. These devices objectively measure physical activity and sedentary behavior by detecting movement and can provide continuous, real-time data. 

Reviewer #2: 

Introduction: While there have been nationally representative studies from Thailand conducted previously on the topic, the need for this study specifically focusing Bangkok is not yet clear. Although authors have attempted to explain it, the reason why it is important to study PA and SB in individuals from Bangkok is important should be explained better.

Thanks for the comment. We’ve revised the introduction, particularly the 3rd and 4th paragraphs to provide justification for focusing Bangkok (page 4). This also includes the comments from Editor and Reviewer 1. 

“Bangkok, the capital and most populous city of Thailand, has experienced significant urbanization, potentially leading to negative implications for physical activity [15], and an increased prevalence of NCDs among its residents [16]. It is among the rapidly growing urban centers in Southeast Asia, with a population surpassing 11 million as of 2023 [17], accounting for approximately 16% of the country’s population [18]. The population growth in Bangkok has been remarkable; in 1950, the city was inhabited by a mere 1.4 million people [17]. However, existing epidemiological investigations of physical activity and sedentary behavior have predominantly been conducted at the national level [19-22]. Limited studies have specifically focused on residents of Bangkok, and those available have tended to concentrate on specific aspects of physical activity, such as exercise or transport behavior, rather than encompassing overall physical activity and sedentary behavior [23, 24]. Bangkok's urbanization is part of a global trend. Understanding how urbanization influences physical activity and sedentary behavior in Bangkok can provide insights into similar trends in other urban centers worldwide, particularly relevant for cities in developing countries that are experiencing rapid urbanization.”

We’ve also revised the last paragraph of the introduction (page 5).

“Therefore, this study aims to …. Such findings are critical for informing policy initiatives that promote healthier lifestyles. These insights can help inform and improve public health strategies and policies in urban areas, fostering a more comprehensive and comparative understanding of physical activity and sedentary behavior in urban settings.”

Reviewer #2: Methods: Page 4: 

What was the rationale for categorising individual income based on 12000 Baht? Please provide a reference. 

Thanks for the comment. The rationale for categorizing individual income based on 12000 Baht is derived from the median. 

Reviewer #2: Results: Page 6: 

The numbers don’t add to 8,538. Please check and correct. 

Thanks for the comment. We’ve checked and revised both the text and Fig 1. 

8,538 = 1,892 + 3,345 + 14 + 150 + 3,137.

“There were 8,538 Bangkok residents who were initially screened for this study. According to the inclusion and exclusion criteria, some participants were excluded based on age (either below 18 years or above 80 years of age, n=1,892), for not being present at their homes on the interview dates (n=3,345), for proving invalid data (n=14), or incomplete sociodemographic data (n=150). The final sample size for analysis consisted of 3,137 participants residing in Bangkok (Fig 1).”

Reviewer #2: Discussion: Page 7: 

While I agree that financial incentives have the potential to influence individual behaviour, it might be argued that it is not a cost-effective approach. Could you suggest better alternatives such as changes to the workstations (e.g., sit-to-stand desks) that are cost-efficient? Use of stairs instead of escalators? 

Thanks for the comment. This is in line with the Editor and Reviewer 1’s comments. We’ve provided examples of sedentary-breaking interventions in the 2nd paragraph of the discussion (pages 15-16).

“…Interventions could be developed to replace sedentary behavior with light-to-moderate-intensity movement several times a day. Examples include using screen time control measures such as electronic lock-out systems to television, computer or smartphone [34]…”

We’ve also provided examples of sedentary-breaking interventions in the 4th paragraph of the discussion (pages 17-18).

“…Examples include using sit-stand desks, treadmill desks, or cycling desks combined with information, counseling, and short break or walking strategies [52]. Implementing point-of-choice prompting software along with educational initiatives may also prove effective [52]…”

Reviewer #2: Discussion: Page 8: 

Why could covid-19 changes have impacted PA and SB in men compared to women?

Thanks for the comment. We were unable to find evidence to answer the question of why COVID-19 changes have impacted physical activity and sedentary behavior differently in men compared to women. Our assumption is that it is related to the decrease in mobility due to social distancing measures. Therefore, we have added this information to the third paragraph on page 17 and removed the clause about men at the end of this paragraph.

“…These measures may have the potential to influence individuals’ physical mobility, with an 11.0% decrease in trips from 9,580 million in 2020 to 8,522 million in 2021 [42]. This could result in a decrease in physical activity and an increase in sedentary behavior [13, 43]”

Reviewer #2: Tables: 

Table 2 need correction: please remove the bullets and format the text in sentence case. 

Thanks for the comment. We’ve revised Table 1.

Reviewer #2: English and grammar: 

The use of English was poor at several places and should be significantly improved. I suggest the manuscript be reviewed by a proficient English speaker. 

Thanks for the comment. The manuscript has been proofread and revised by a proficient English speaker as also suggested by the Editor and Reviewer 1.

---

## [Decision Letter · Decision Letter 1]

18 Sep 2023

Patterns and correlates of physical activity and sedentary behavior among Bangkok residents: A cross-sectional study

PONE-D-23-18854R1

Dear Dr. Topothai,

We’re pleased to inform you that your manuscript has been judged scientifically suitable for publication and will be formally accepted for publication once it meets all outstanding technical requirements.

Kind regards,

Nipun Shrestha, Ph.D.

Academic Editor

PLOS ONE

Additional Editor Comments (optional):

Reviewers' comments:

Reviewer's Responses to Questions

**Comments to the Author**

1. If the authors have adequately addressed your comments raised in a previous round of review and you feel that this manuscript is now acceptable for publication, you may indicate that here to bypass the “Comments to the Author” section, enter your conflict of interest statement in the “Confidential to Editor” section, and submit your "Accept" recommendation.

Reviewer #1: All comments have been addressed

Reviewer #2: (No Response)

2. Is the manuscript technically sound, and do the data support the conclusions?

Reviewer #1: Yes

Reviewer #2: Yes

3. Has the statistical analysis been performed appropriately and rigorously? 

Reviewer #1: Yes

Reviewer #2: Yes

4. Have the authors made all data underlying the findings in their manuscript fully available?

Reviewer #1: Yes

Reviewer #2: No

5. Is the manuscript presented in an intelligible fashion and written in standard English?

Reviewer #1: Yes

Reviewer #2: Yes

6. Review Comments to the Author

Reviewer #1: The authors have satisfactorily addressed my research concerns. Congratulations to the authors' work, which may be helpful to understand the PA pattern in the low and middle income country. Good work.

Reviewer #2: I am thankful to the authors for revising the manuscript. I can now see that the manuscript has been adequately revised and addresses my comments.

7. PLOS authors have the option to publish the peer review history of their article (what does this mean?). If published, this will include your full peer review and any attached files.

Reviewer #1: No

Reviewer #2: No

---

## [Editor Report · Acceptance letter]

25 Sep 2023

PONE-D-23-18854R1 

Patterns and correlates of physical activity and sedentary behavior among Bangkok residents: A cross-sectional study 

Dear Dr. Topothai:

I'm pleased to inform you that your manuscript has been deemed suitable for publication in PLOS ONE. Congratulations! Your manuscript is now with our production department. 

Kind regards, 

on behalf of

Dr. Nipun Shrestha 

Academic Editor

PLOS ONE